# Involvement of CONSTANS-like Proteins in Plant Flowering and Abiotic Stress Response

**DOI:** 10.3390/ijms242316585

**Published:** 2023-11-22

**Authors:** Bingqian Zhang, Minghui Feng, Jun Zhang, Zhangqiang Song

**Affiliations:** 1Key Laboratory of Cotton Breeding and Cultivation in Huang-Huai-Hai Plain of Ministry of Agriculture and Rural Affairs, Institute of Industrial Crops, Shandong Academy of Agricultural Sciences, Jinan 250100, China; qqniuliji@gmail.com (B.Z.); 19811807935@163.com (M.F.); scrczj@saas.ac.cn (J.Z.); 2College of Life Science, Shandong Normal University, Jinan 250358, China

**Keywords:** CONSTANS-like, transcription complex, photoperiod, flowering time, abiotic stress

## Abstract

The process of flowering in plants is a pivotal stage in their life cycle, and the CONSTANS-like (COL) protein family, known for its photoperiod sensing ability, plays a crucial role in regulating plant flowering. Over the past two decades, homologous genes of *COL* have been identified in various plant species, leading to significant advancements in comprehending their involvement in the flowering pathway and response to abiotic stress. This article presents novel research progress on the structural aspects of COL proteins and their regulatory patterns within transcription complexes. Additionally, we reviewed recent information about their participation in flowering and abiotic stress response, aiming to provide a more comprehensive understanding of the functions of COL proteins.

## 1. Introduction

The precise coordination of specific developmental processes and appropriate times is crucial for plant survival and reproduction. The timing of flowering, for instance, enables plants to allocate sufficient time for optimal seed development and maturation prior to the onset of winter. In addition, the plant’s ability to respond to photoperiod also enables it to anticipate annual environmental changes and proactively adapt accordingly. This requires plants to possess special mechanisms that enable them to sense seasonal differences by detecting and responding to changes in photoperiod [1]. There are three types of photoperiodic response plants: long-day plants (LDPs), the response of which is induced when the photoperiod surpasses the critical day length (CDL); short-day plants (SDPs), the response of which is induced when the photoperiod is shorter than the CDL; and day-neutral plants (DNPs) that do not respond to photoperiod [2]. The time measurement in plants is achieved through an endogenous time-keeping mechanism known as the circadian rhythm, wherein plants perceive light signals via photoreceptors and regulate the rhythmic expression of *CO* through circadian components [1,3,4,5]. As a representative member of the COL protein family, CO plays a key role in the regulation of photoperiodic flowering, and is specifically regulated by time and space [6]. Research on the CO domain reveals functional support for the regulatory mechanism of CO transcription factors, and the identification of CO regulatory factors at the transcriptional and post-transcriptional levels enriches our knowledge of the CO-FT pathway [3,7,8].

The first COL protein found in plants was AtCO of Arabidopsis, which is able to promote plant flowering under long-day conditions [9]. Later, researchers located and cloned *HEADING DATE 1* (*HD1*) in rice, which showed significant amino acid sequence similarity to At*CO* [10]. Further research revealed that HD1 plays a similar role in the flowering pathway to that of CO in Arabidopsis and has a similar function, confirming that *HD1* is a homologous gene of *AtCO* in rice (*Oryza sativa*) [11,12]. Referring to the sequences of *CO* and *HD1*, *CO/COL* family genes have been identified sequentially in various plants. Among dicotyledonous plants, 17 members were identified in Arabidopsis (*Arabidopsis thaliana*) [13], 13 members in tomato (*Solanum lycopersicum*) [14], 20 members in radish (*Raphanus sativus*) [15], 10 members in sugar beet (*Beta vulgaris* ssp. *Vulgaris*) [16], 11 members each in alfalfa (*Medicago truncatula*) [17] and soybean (*Glycine max* [L.] Merr.) [18], and 25 members each in rape (*Brassica rapa*) [19] and banana (*Musa paradisiaca*) [20]; meanwhile, among monocotyledonous plants, 16 members have been identified in rice (*Oryza sativa*) [21], 19 members in maize (*Zea mays*) [22], and 9 members in barley (*Hordeum vulgare* L.) [21].

The discovery of these COL members has enhanced our understanding of how they participate in the regulation of flowering based on photoperiods. The research has shown that some COL members may function through interfering with the CO complex [23,24]. Moreover, many *COL* members show significant changes in expression abundance in response to external stressors [25,26]. Exploring the mechanisms of *COL* genes responding to stress is important in order to understand how plants coordinate growth and development, especially in terms of flowering under stress conditions [27,28]. This review aims to provide an overview of the functional role of COL members in photoperiod regulation and in abiotic stress response as reported in recent years.

## 2. Function of COL Protein Domains

The COL proteins typically comprise an N-terminal B-box domain and a C-terminal CCT domain, along with a central region containing glutamine-rich sequences [9]. Based on their conserved B-box and CCT domains, they can be categorized into three subfamilies [13]: group I proteins includes two B-box domains, one CCT domain, and one VP motif; group II proteins has one B-box domain and one CCT domain; and group III proteins has one B-box domain, a divergent zinc finger structure, and a CCT motif.

The B-box domain in COL members is highly conserved, characterized by a single motif consisting of 30–40 amino acid residues that bind zinc ions through Cys, Asp, and His residues to form a zinc finger tertiary structure [29]. In vitro experiments, such as GST pull-down assays, have demonstrated that the B-box domains can interact with each other, forming diverse oligomeric states, predominantly tetramers [24,30]. It has been reported that BBX proteins like AtBBX19, 28, 30, and 31 can physically associate with AtCO via their B-box domains, indicating that COL proteins form oligomeric complexes that function not only through self-interaction but also heterologous interaction of B-box domains [31,32,33].

Proteins are targeted to the nucleus via nuclear localization signal (NLS) sequences [34]. A conserved bipartite NLS (consisting of RK-X11-R sequence) is present in the CCT domain at the C terminus of COL proteins, facilitating their nuclear import [35,36]. Studies on AtCO suggest that the CCT domain plays an essential role in binding downstream motifs [37]. CCT domain can also participate in protein–protein interactions [38].

The central region between the B-box and CCT domains in most COL proteins contains uninterrupted glutamine-rich sequences [39]. Following truncation of the *CO* gene, it was revealed that the intermediate fragments exhibited transcription activation capability, while the intact protein containing the middle region exhibited a higher transcriptional activation capacity than the truncated protein [37]. Consequently, it is widely postulated that the transcriptional activation capacity of COL protein predominantly originates from this specific region.

## 3. Assembly of COL Transcription Complex

The CCT domain of CO protein exhibits structural similarity to the NF-YA subunit of NUCLEAR FACTOR-Y A (NF-YA) [38]. Within the trimeric complex formed by NF-YA/YB/YC, NF-YA interacts with the NF-YB/YC dimer through an α-helix [24,40]. This trimeric complex demonstrates specificity in binding to CCAAT sequences [41]. Previous studies postulated that CO protein competes with NF-YA for occupancy of the NF-YB/NF-YC dimer, thereby exerting its transcriptional function through recognizing CCAAT motifs in promoter sequences of downstream target genes [41,42,43].

Currently, research has demonstrated that the formation of NF-YB/NF-YC dimer depends on their HFD (histone fold domain), which facilitates a responsive electrostatic-interaction network, thereby enabling the dimer to adaptively trimerize with various interaction partners [44]. The NF-YB/NF-YC dimer itself does not have DNA binding specificity. The resemblance between CCT and NF-YA enables NF-YB/YC to acts as a scaffold for CCT to bind to DNA [45,46,47]. This interaction is facilitated by the first α-helix of CCT, while sequence-specific binding relies on the second α-helix [30]. As a downstream target gene of CO and several COL members, *FLOWERING LOCUS T* (*FT*) has been experimentally shown to harbor CORE1 (TGTGA, −220 bp), CORE2 (TGTGG, −161 bp), and P1/P2 (CCACA/TGTGG, −267 bp/−285 bp) motifs in its promoter, which serve as direct binding sites for CO-CCT complex and, collectively, constitute the CO response region (CORR) of *FT* promoter [37,48]. The spatial interference (18 bp) hinders the simultaneous binding of P1 and P2.

In order to define the scope of influence of CO on the promoter of *FT* (*pFT*), β-glucuronidase (GUS) reporter gene analysis was performed and the −5.7 kb upstream sequence of *FT* was identified as the minimal functional length, which contains three possible CO regulatory regions [48]. Although CCAAT does not appear to serve as a binding site for CO itself, the CCAAT motif at −5.3 kb of the *FT* promoter in cruciferous plants is relatively conserved [48,49]. Moreover, chromatin conformation capture (3C) has revealed a dynamic interaction between CCAAT (−5.3 kb) and CORE regions mediated by NF-Y [50]. This finding demonstrates that CCAAT likely acts as a crucial cis-regulatory element facilitating sequential interaction between CO and *pFT*.

Based on various studies, researchers have proposed a “recruitment” model wherein the NFY complex modulates the DNA structure by binding to conserved regulatory regions within *pFT*, thereby establishing time-specific chromatin loop structures that facilitate recruitment of CO through interactions between NF-Y and CO-CCT at the proximal *pFT* region (Figure 1) [48,50]. Additionally, a “silencing relief” model has been proposed, wherein the binding of POLYCOMB GROUP (PcG) proteins to *pFT* effectively suppresses *FT* expression. The interaction between the NF-CO and NF-Y complexes synergistically relieves PcG repression and initiates transcription of *FT* [51]. However, research has revealed that NF-YA and CO compete for occupancy at the same NF-YB/YC HFD dimer binding site, suggesting the potential involvement of unidentified proteins in mediating communication between NF-YA and CO during this process [30].

Considering the emergence of the CO transcription complex, it is conceivable that COL members sharing similar motifs might participate in this process. Previous investigations demonstrated that certain COL members can modulate *FT* transcription via physically interacting with CO and depleting its transcriptional activity [52,53]. The CO transcription complex appears to function as a signal center, accommodating the inclusion of COL members to incorporate additional information. However, numerous COL members, despite possessing CCT domains, do not directly target *pFT* or interfere with CO complexes [54,55,56]. The HD1 protein strictly recognizes the TGTGG motif, in contrast to AtCO, which conservatively recognizes only the four-base sequence TGTG [57]. The specificity of CO/COL in recognizing downstream targets may also determine the differences in their function across different species, providing some theoretical support for precise flowering regulation. Studies conducted using complex models, although not without flaws, have significantly enhanced our comprehension of CO regulatory mechanisms and provide indispensable theoretical support for the precise regulation of flowering processes.

## 4. COL Involved in Photoperiod Flowering

### 4.1. Regulation of CO in Photoperiod Flowering

CO responds to photoperiodic changes and modulates the expression of the flowering gene *FT* [9,58]. The CO-FT photoperiod pathway has been identified to be conserved across multiple species and has been studied carefully in the long-day plant Arabidopsis [59,60]. In this pathway, the core regulatory factor CO is regulated at transcriptional and post-transcriptional levels. In transcriptional regulation, GIGANTEA (GI), FLAVIN-BINDING-KELCH REPEAT-F-BOX 1 (FKF1), and CYCLING DOF FACTOR 1 (CDF1) are key regulators of *CO* transcription [61,62]. In post-transcriptional regulation, CONSTITUTIVELY PHOTOMORPHOGENIC 1 (COP1), HIGH EXPRESSION OF OSMOTICALLY RESPONSIVE GENE 1 (HOS1), FKBP12 (FK506 BINDING PROTEIN 12 kDa), and PHYTOCHROMES A/B (PHYA/B), CRYPTOCHROME 2 (CRY2) are directly or indirectly involved in the regulation of *CO* [63,64,65,66,67]. During the whole process, the input of external light information comes from the red-light receptor phytochrome PHYA/B and blue-light receptor FKF1/CRY. The CO photoperiodic flowering pathway in Arabidopsis is shown in Figure 2.

In the morning, the CDF1 transcription factor inhibits *CO* transcription by binding to DOF binding sites in conjunction with TOPLESS (TPL) co-repressor [62]. Later, FKF1/ZEITLUPE (ZTL) is stabilized by blue light and interacts with GI-HSP90 to alleviate CDF-mediated inhibition of *CO* [68]. The high expression of FKBP12 protein in the morning can mitigate COP1-SPA(SUPPRESSOR OF PHYA-105) degradation of CO through its interaction with the CO-CCT domain [65]. Additionally, BBXs (BBX19, 30, 31), TARGET OF EAT1/2 (TOE1/2), PHYB, HOS1, and DELLA negatively regulate CO protein activity or abundance in the morning to ensure that CO does not accumulate too early and initiate downstream *FT* expression to prevent premature flowering in Arabidopsis [31,33,69,70,71,72].

In the afternoon, *CO* transcription is upregulated by the positive regulatory factors FLOWERING BHLH (FBH) and TEOSINTE BRANCHED 1/CYCLOIDEA/PROLIFERATING CELL NUCLEAR ANTIGEN FACTOR (TCP). PHYTOCHROME AND FLOWERING TIME 1 (PFT1) acts as an intermediary between FBH and TCP proteins, while specific members of TCP, such as TCP4, are activated by GIGANTEA (GI) to stimulate *CO* transcription [73,74]. PHYA, FKF1, and CRY2 all positively stabilize CO protein through inhibiting COP1 function, resulting in gradual accumulation of CO protein and initiation of *FT* transcription [67,75,76]. The NF-Y complex enhances the binding affinity between CO protein and *FT* promoter [37]. Additionally, CO also induces the transcription of *SODIUM POTASSIUM ROOT DEFECTIVE 1* (*NaKR1*) gene encoding FT transport protein [77]. Consequently, both CO and FT exhibit peak abundance later in the day.

Despite the reduction in CDF1 and CDF2 abundance by GI-FKF1 at nighttime, a sufficient amount of CDF protein remains to inhibit *CO* transcription [78]. The nocturnal peak expression of *ABI5-BINDING PROTEIN2* (*AFP2*) gene facilitates its interaction with CO through its C-terminus, while also interacting with the N-terminus of TOPLESS-RELATED PROTEIN2 (TPR2) to form a CO-AFP2-TPR2 complex [79]. This intricate complex orchestrates CO degradation and concurrently inhibits *FT* transcription. Moreover, stable factors like FKBP12 are depleted at night, leading to COP1-SPA E3 ubiquitin ligase-mediated ubiquitination and subsequent degradation of CO protein [65].

### 4.2. Other COL Members Participate in Photoperiod Flowering

A substantial body of genetic and biochemical research has demonstrated that, apart from CO, the COL family encompasses multiple members that actively or negatively regulate flowering through distinct molecular mechanisms (Table 1). Overexpression of *AtCOL1* in Arabidopsis can expedite the circadian clock and abbreviate both long-day and short-day rhythms [80]. Overexpression of *AtCOL2* induces a delay in flowering, albeit with relatively minor and unstable effects [80]. Through interacting with BBX32 protein via its B-box domain, AtCOL3 facilitates binding to the promoter region of the downstream target gene *FT*, thereby repressing transcription of the *FT* gene and retarding flowering under long-day (LD) conditions [81]. As a suppressor of flowering, AtCOL4’s mode of action is influenced by day length; it acts upstream of *FT* and *SUPPRESSOR OF OVEREXPRESSION OF CO 1 *(*SOC1*) under LD conditions [82]. Overexpression of *AtCOL5* can induce flowering in Arabidopsis under short-day (SD) conditions, while downregulation of its expression does not directly impact the flowering phenotype, suggesting a potential redundancy in COL5 function. It is noteworthy that evolutionarily, AtCOL5 may have preceded CO and served as its precursor [83]. Interactions between AtCOL6 and AtCOL16 (BBX15)/AtCOL7 (BBX16) protein members have been reported to delay flowering by inhibiting *FT* transcription mediated by CO [84]. The role of AtCOL7 in flower regulation remains undiscovered; however, research on this protein suggests a possible light signal transduction pathway, where it connects external light information with downstream *SUPERROOT2* (*SUR2*) gene transcription activation through upstream PHYB to coordinate auxin levels [85]. AtCOL8 and AtCOL12 interact with CO to inhibit the transcriptional activation of *FT* expression, and their degradation is dependent on COP1 [52,53]. AtCOL9 directly suppresses the transcriptional expression of *CO*, and itself is regulated by the biological clock [86]. The potential impact of AtCOL10 on flowering regulation in Arabidopsis remains unexplored; however, its homologous protein CmBBX8 (CmCOL10) in chrysanthemum promotes flowering by accelerating the transcription of *TERMINAL FLOWER 1* (*CmFTL1*) [87]. AtCOL11 can induce the production of JASMONATE (JA) and may be involved in JA-mediated flower organ development [88]. AtCOL13 plays an important role in plant photomorphogenesis [89,90,91].

Furthermore, COL members from exogenous genetic backgrounds can also influence flowering regulation in Arabidopsis. Transformation experiments with exogenous genes revealed that multiple COL members from mango (*Mangifera indica* L.; MiCOL1s, MiCOL2s, MiCOL9s, and MiCOL16s) suppress flowering by affecting the expression of *AtFT*, while PvCO1 in bamboo (*Phyllostachys violascens*) and VrCOL2 in mungbean (*Vigna radiata*) also reduce the expression of *AtFT* in Arabidopsis, leading to delayed flowering [92,93,94,95,96].

**Table 1 ijms-24-16585-t001:** Role of Arabidopsis COL in plant flowering and other regulation (ND, not determined).

Gene Name	BBX Name	Gene Locus ID	Biological Process	Function	Target Genes	Interacting Proteins	References
*CO*	BBX1	AT5G15840	Flowering	Positive	*FT*	COP1, HOS1	[64,97]
						TOE, DELLA	[69,72]
						AFP2, FKBP12	[65,79]
						BBX19, BBX30	[31,33]
						BBX31	[5]
*COL1*	BBX2	AT5G15850	Flowering	Positive	*FT*	ND	[80]
*COL2*	BBX3	AT3G02380	Flowering	Negative	ND	ND	[80]
*COL3*	BBX4	AT2G24790	Flowering	Negative	*FT*	BBX32, COP1	[81,98]
						PIF3, PHYB	[99]
*COL4*	BBX5	AT5G24930	Flowering	Negative	*FT*	CO	[82]
*COL5*	BBX6	AT5G57660	Flowering	Negative	*FT*	ND	[83]
*COL6*	BBX14	AT1G68520	Flowering	Negative	*FT*	CO	[84]
*COL7*	BBX16	AT1G73870	Branching	Positive	*SUR2*		[85]
			Flowering	Negative	*FT*	CO	[84]
*COL8*	BBX17	AT1G49130	Flowering	Negative	*FT*	CO	[52]
*COL9*	BBX7	AT3G07650	Flowering	Negative	*CO*, *FT*	ND	[86]
*COL10*	BBX8	AT5G48250	ND	ND	ND	ND	
*COL11*	BBX9	AT4G15250	Flowering	Positive	ND	ND	[88]
*COL12*	BBX10	AT3G21880	Flowering	Positive	*FT*	CO, COP1, SPA1	[53]
*COL13*	BBX11	AT2G47890	Photomorphogenesis	Positive	*CHLH*, *HEMA1*	PHYB, PIF4	[89,91]
*COL14*	BBX12	AT2G33500	ND	ND	ND	ND	
*COL15*	BBX13	AT1G28050	ND	ND	ND	HAPs	[38]
*COL16*	BBX15	AT1G25440	Flowering	Negative	*FT*	CO	[84]

### 4.3. Functional Role of COL in LDPs, SDPs, and NDPs

The plant photoperiod, which was discovered in the last century, is defined as the ability of plants to recognize changes in day length and make adjustments to flower at the appropriate time [100]. Plans are divided into long-day plants (LDPs), short-day plants (SDPs), and day-neutral plants (NDPs) based on differences in their response to photoperiod [2]. The peak expression of *CO* in Arabidopsis was observed only in the afternoon under long-day conditions. Regulation at the transcriptional and post-transcriptional levels of *CO* requires sufficient sunlight time for its function. Otherwise, the abundance of CO is not enough to activate the expression of *FT*, that is, Arabidopsis flowering is delayed under short-day conditions [101].

As a short-day plant, rice (*Oryza sativa*) has a flowering pathway similar to that in Arabidopsis. *OSGI* (*GI*), *Hd1* (*CO*), *Hd3a* (*FT*), and *RFT1* (*FT*) in rice correspond to the homologous genes *GI*, *CO*, and *FT*, respectively, in Arabidopsis [10,102,103]. What is not identical is that there are two pathways regulating flowering in rice [104,105]. In the OSGI-Hd1-Hd3a pathway of rice, similar to Arabidopsis, *Hd1* is transcriptionally regulated by upstream OSGI and reaches its peak at dusk (LD) or at night (SD) [106]. Notably, Hd1 (CO) has dual functions, and its function changes under the influence of phytochrome; it converts into an inhibitor of *Hd3a* during the day, so it only promotes the expression of *Hd3a* at night [107,108]. The properties of Hd1 prompt rice to recognize short-day photoperiods and initiate the flowering process. In addition, there is a unique long-day suppression pathway in rice, the GRAIN NUMBER, PLANT HEIGHT AND HEADING DATE 7 (Ghd7)-EARLY HEADING DATE 1 (Ehd1)-Hd3a/RFT1 pathway [109]. Ghd7 and Ehd1 are new photoperiod regulators that have emerged in rice evolution. Ehd1 promotes flowering by promoting downstream *Hd3a* and *RFT1* expression. *Ghd7* is expressed under long-day conditions and inhibits the transcription of *Ehd1*, inhibiting flowering under LD conditions [110,111,112].

There are many possible reasons for plant photoperiod insensitivity. In day-neutral plants, such as roses (*R. chinensis*), *RcCO* is highly expressed only under long-day conditions and promotes flowering by regulating the expression of downstream *RcFT*. However, under short days, rose flowering was not significantly affected. RcCOL4, one of the Rose COL members, can enhance the ability of RcCO to bind *RcFT* promoter in SD by physically interacting [113]. In tomato (*Solanum lycopersicum* L.), although COL members *SlCOL*, *SlCOL4a*, and *SlCOL4b* were identified as potential flowering promoters, it is unclear whether they determine photoperiod insensitivity [14]. Tomato flowering inhibitor SELF PRUNING 5G (SP5G) can inhibit the expression of tomato *FT* (*SFT*). In wild varieties, *SP5G* is highly expressed under long days, but due to cis-regulatory variation, expression is lower in cultivated tomatoes, which may explain the genetic changes in tomatoes from SDP to NDP [114]. In cucumber (*Cucumis sativus*), the upstream region of the *FT* (*CsFT*) gene differs among cultivars grown at different latitudes. The lack of photoperiod sensitivity may result from the loss of upstream regulatory elements of *FT* after long-term artificial selection, resulting in a lack of precise control of *CsFT* expression [115].

## 5. COL Is Involved in Abiotic Stress Response

### 5.1. CO-FT Pathways and Abiotic Stress

The response of flowering to environmental signals has long been a subject of significant interest. Under unfavorable circumstances, plants possess the ability to regulate their flowering time by either advancing or delaying it, thereby optimizing seed survival [116,117]. Although the mechanisms underlying the integration of flowering and stress responses may be highly intricate, numerous protein members within the flowering pathway have demonstrated associations with external stress [118]. The GI-CO-FT pathway in leaves can respond to diverse stress signals, while shoot apical meristem (SAM) differentiation is influenced by various environmental cues (Figure 3).

Stress signals such as drought, salt, and cold can modulate the abundance of GI and thereby influence the initiation of the flowering pathway. This highlights the multifunctionality of GI as a central signaling hub [28,119,120]. Drought stress can upregulate *GI* transcription, and by promoting early flowering through the GI-CO-FT pathway but not solely relying on it, GI can also directly bind to the *pFT* without CO [121,122]. Previous studies demonstrated that cold stress induces independent expression of *GI*, unrelated to C-REPEAT/DREB BINDING FACTOR (CBF) [123]. Subsequent studies found that GI can participate in the regulation of various processes through CDF, and enhances cold tolerance by releasing CDF [124]. GI is also considered a pivotal component of the salt stress adaptation pathway. GI and SALT OVERLY SENSITIVE (SOS2) kinases form protein complexes, in which, under salt stress conditions, GI undergoes degradation by 26S proteasome, leading to the release of SOS2. Subsequently, SOS2 activates other Na^+^/H^+^ antiporters (SOS1, SOS3) to maintain cellular homeostasis [125,126,127,128].

EARLY FLOWERING3 (ELF3) acts as a constituent of the ELF3-ELF4-LUX ARRHYTHMO (LUX) complex, governing light input to the circadian clock in the flowering pathway [129]. ELF3 potentially participates in GI-SOS2 by modulating GI levels, likely accomplished through COP1-ELF3 complex-mediated degradation of GI protein [130]. Moreover, serving as an upstream regulator of PIF4, a pivotal regulatory factor for abiotic stress tolerance, ELF3 controls *PIF4* transcription, thereby enhancing plant salt tolerance [131,132,133]. CO is regulated not only by GI, but also by the protein HIGH EXPRESSION OF OSMOTICALLY RESPONSIVE GENES1 (HOS1), leading to degradation under cold stress conditions [134]. SHORT VEGETATIVE PHASE (SVP), a crucial regulatory factor in flower development, acts as a negative regulator of flowering upstream of *FT* and *TWIN SISTER OF FT* (*TSF*) [135]. Drought stress induces upregulation of *SVP*, which subsequently triggers downstream AtBG1 protein expression to facilitate ABA accumulation and enhance plant drought tolerance [136]. Cold stress can enhance the expression of CBF members, which in turn activate the transcription of *FLOWERING LOCUS C* (*FLC*) [137]. Under cold stress conditions, FLC acts as a repressor of two key flowering pathway integrators, FT and SUPPRESSOR OF OVEREXPRESSION OF CO1 (SOC1), thereby causing a delay in flowering time [138]. Upon relief from cold signaling, SOC1, GI, and other factors reciprocally suppress the CBF-mediated cold response pathway [139].

The function of the CO-activated FT protein is mediated through its transport via the phloem to the shoot apical meristem (SAM) [140]. Subsequently, FT requires assistance from a bZIP protein called FD for its flowering-promoting activity [141,142]. Upon FD induction, SOC1, the downstream protein of FT, can activate *LEAFY* (*LFY*), a floral meristem identity gene [143]. LFY and other floral formation factors initiate flower development at the primordium in the SAM [144,145]. The Arabidopsis flower inhibitory factor BROTHER OF FT AND TFL1 (BFT) is induced by high salinity and delays flowering by competitively binding with FD-FT to disrupt FT’s function [146]. ABA-RESPONSIVE ELEMENT-BINDING FACTOR (ABF), a bZIP transcription factor, plays a crucial role in ABA signal transduction during drought and osmotic stress [147]. Drought induces an increase in ABA content, and ABA-dependent ABF members (ABF3 and ABF4) directly promote early flowering and a drought-escape (DE) response via transcriptional regulation of *SOC1* [148].

### 5.2. COL Participates in Abiotic Stress Response

The COL family governs the photoperiodic flowering pathway, and numerous studies have reported the responses of its members to abiotic stress (Figure 4). A study on the significant induction of *MaCOL1* expression in response to cold and pathogen infection stress in banana (*Musa acuminate*) may be the first report of COL members involved in abiotic stress (2012) [26].

In model plant Arabidopsis, AtCOL4 participates in ABA pathway and salt-stress responses through an ABA-dependent signaling pathway to positively regulate non-biological stress tolerance [149]. Conversely, AtCO negatively mediates salt tolerance in Arabidopsis by interacting with four ABSCISIC ACID-RESPONSIVE ELEMENT BINDING FACTORS (ABF1, ABF2, ABF3, and ABF4) [150]. Additionally, CO is also a JAZ-binding factor that participates in jasmonic acid signal transduction through protein interactions [151]. OMG1, an uncharacterized Arabidopsis III class COL protein, functions as a regulator of the ROS pathway by modulating the expression of ROS pathway-related genes *MYB77* and *GRX480* [152]. The blue light receptor CRYPTOCHROME 2 (CRY2) stabilizes at low temperatures and inhibits COP1-mediated degradation of LONG HYPOCOTYL 5 (HY5) through its interaction with COP1. As a downstream target gene of HY5, *AtCOL10* positively regulates cold tolerance by modulating the expression of cold-responsive *COR* (cold-regulated) genes [153].

In addition, overexpression of *Ghd2* in rice (*Oryza sativa*) significantly impairs drought resistance and plays a crucial role in drought-induced leaf senescence [154]. In the halophyte *Tamarix hispida*, ThCOL2 protein enhances the activity of protective enzymes, reduces ROS and MDA accumulation in plants, and mitigates cell damage to enhance salt tolerance in transgenic plants [155]. In maize (*Zea mays*), eight members of group I ZmCOL possess ABA-responsive cis-elements, and their expression levels are modulated by ABA, indicating their widespread involvement in ABA response [25]. BnCOL2 in rapeseed (*Brassica napus*) regulates plant tolerance to drought stress by modulating ABA response and regulating the expression of drought-related genes. Heterologous overexpression of *BnCOL2* significantly compromises drought tolerance in Arabidopsis under drought stress conditions [156]. Overexpression of mango (*Mangifera indica* L.) *COL* members *MiCOL16A* and *MiCOl16B* in Arabidopsis enhances the salt tolerance and drought resistance [94]. In soybean (*Glycine max* [L.] Merr.), GmCOL1a has been found to enhance salt tolerance through promoting the expression of salt transport-related proteins, thereby effectively reducing the Na+/K+ ratio in plants. Moreover, GmCOL1a positively regulates the transcription of *GmLEA* and *GmP5CS* to improve drought resistance in soybeans [157]. In apple (*Malus × domestica*), MdCOL9 is regulated by ubiquitination through its interaction with the drought-responsive protein MdMIEL1. This interaction activates downstream positive regulatory factors such as *MdERF1*, *MdGLK1*, and *MdERD15* to enhance drought resistance [158].

Furthermore, several studies have reported dynamic transcriptional changes in *COL* genes under diverse abiotic stress conditions. For instance, under drought stress, *GhCOL3*, 4, 14, 16, 17, and 20 in cotton (*Gossypium hirsutum*) were upregulated and under salt stress *GhCOL6*, *8*, *9*, *10*, *11*, *13*, *18*, *19*, *21* were downregulated [25]. In response to high-temperature stress, *CaCOL1*, *6*, *7*, *8*, and *9* in pepper (*Capsicum annuum*) were upregulated while *CaCOL2*, *3*, and *5* were downregulated. Similarly, under osmotic stress conditions, the expression levels of *CaCOL2*, *3*, *7,* and *8* were observed to be upregulated following 2 h treatment [159]. In sunflower (*Helianthus annuus* L.), under drought stress, the expression of *HaCOL3* was significantly downregulated, while the *HaCOL19* and *HaCOL22* were significantly upregulated [160]. Although the functions and molecular mechanisms of these COLs have not been investigated, promoter element analysis or similar methods can be used to quickly explore the possible upstream members. For example, through yeast one-hybrid library screening, it was found that ABA-induced BrABF3 can directly activate *BrCO* transcription [161]. These investigations will serve as valuable references for further elucidating the mechanisms of COL proteins responding to non-biological stress and exploring the intricate relationship between flowering regulation and stress response.

## 6. Future Perspectives

The transition from vegetative growth to reproductive growth is a pivotal event in plant development, where flowering serves to initiate reproductive growth. Plants possess the ability to perceive light and regulate the onset of flowering. This study provides a comprehensive understanding about the role and molecular mechanisms underlying COL proteins in governing flowering regulation, although many intricacies remain undisclosed. The assembly and regulatory mechanism of the COL transcriptional complex binding to the *FT* promoter needs to be further elucidated. Currently, the conjunction of NY-F complex with CO requires the participation of some unknown members. Uncovering the intrinsic characteristics of COLs and identifying these key factors remain subjects of research. COL actively participates throughout the entire flowering process beyond merely activating/inhibiting *FT* transcription regulatory functions. The fact that multiple COL members can participate in the regulation of plant photoperiodic flowering at multiple pathways highlights the intricate complexity of COL regulation mechanisms.

Investigating the COL family is important, not only to elucidate the mechanisms underlying plant growth and development, but also to enhance crop yields. As a pivotal gene family governing flowering through photoperiodic regulation, the COL family can be regarded as having a central role in integrating signals during light-mediated flowering in plants. In agricultural production, the timing of flowering directly affects crop yields. The *COL* genes have significant potential in enhancing crop productivity. In crops, various *COL* genes exhibit pleiotropic effects on traits associated with yield, plant height, and resistance against environmental stresses. For instance, *OsCOL9*, *OsCOL10*, and *OsCOL16* have the capacity to positively modulate characteristics including grain count per panicle, length of inflorescence, and overall plant stature, resulting in improved rice grain output irrespective of photoperiodic constraints [162,163,164]. In wheat, *TaCOL-B5* overexpression was reported to induce greater tillering and spike formation, resulting in an approximately 12% increase in productivity [165]. Moreover, it has been reported that all eight *ThCOL* genes in *Tamarix hispida* exhibit responsiveness to stress [155]. Even though numerous *COL* genes across various species have been identified and characterized, several members of the COL family have not yet been extensively investigated, and their biological functions and molecular mechanisms remain poorly elucidated. Currently, the COL protein shows potential as an integrated regulator of flowering, stress response, and yield formation. But how COL proteins integrate various environmental and developmental signals to coordinate the appropriate plant response will be a fascinating topic.

## Figures and Tables

**Figure 1 ijms-24-16585-f001:**
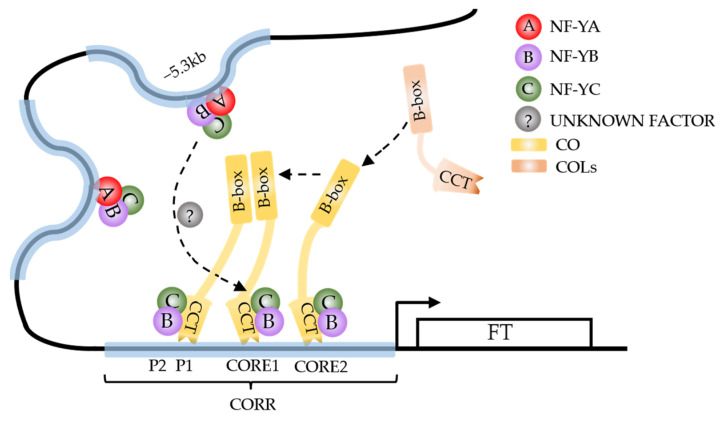
CONSTANS (CO) transcription complex “recruitment” model. NY-FB and NF-YC physically bind to CO through the CCT domain and provide a scaffold to assist CO binding to the CORR region of *pFT*. NY-FA/YB/YC trimeric complexes can bind to multiple sites on *pFT* to promote the formation of chromatin loops at this specific sites, facilitating its proximity to the CORR motif, and ultimately promoting the binding of NF-YB/YC/CO complex to CORR elements through the function of unknown factor (long dashed arrow). Other COL members may participate in the assembly of CO complexes through interactions between B-box domains (short dashed arrow). The solid arrow represents the promotion of *FT* expression by the CO complex.

**Figure 2 ijms-24-16585-f002:**
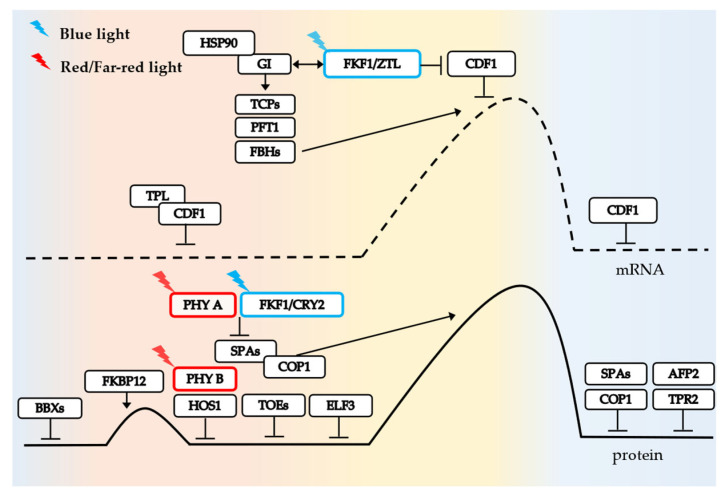
CONSTANS (CO) photoperiodic flowering pathway. Background color represents change from morning to evening during the day (left to right). In the morning, *CO* transcription levels are inhibited by the CDF1-TPL complex. In the afternoon, blue light can stabilize FKF1/ZTL, thereby releasing inhibition of CDF1. Positive transcriptional regulators such as TCP also facilitate transcription. *CO* mRNA levels accumulate to a peak. Similarly, at protein level, CO abundance gradually increases and reaches its highest level in the afternoon, with a small peak in the morning, possibly due to the action of FKBP12. After that, members such as PHYB and HOS1 quickly control CO abundance to avoid its premature accumulation. Until the afternoon, PHYA, FKF1, and CRY2 relieve CO degradation by COP1.

**Figure 3 ijms-24-16585-f003:**
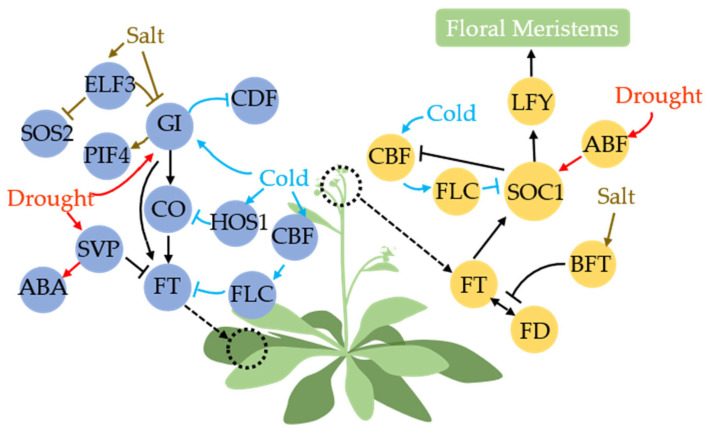
Effects of abiotic stress (drought, cold, salt) on the photoperiodic pathway in Arabidopsis. Blue and yellow circles represent photoperiodic members in leaf and shoot apical meristem (SAM), respectively (these two positions are marked with dashed circles). Black arrows represent transmission of photoperiodic signals. Drought stress signals can directly affect GI, FT, and SOC1 in the photoperiodic pathway through SVP and ABF to promote plant flowering (red arrow). Salt stress signals can promote degradation of GI and interfere with FT function by inducing BFT/TFL1 to hinder plant flowering (brown arrow). Cold stress signals can indirectly block CO, FT, and SOC1 in the photoperiodic pathway to delay plant flowering. Cold can also directly induce GI and release CDF to improve plant cold tolerance (blue arrow).

**Figure 4 ijms-24-16585-f004:**
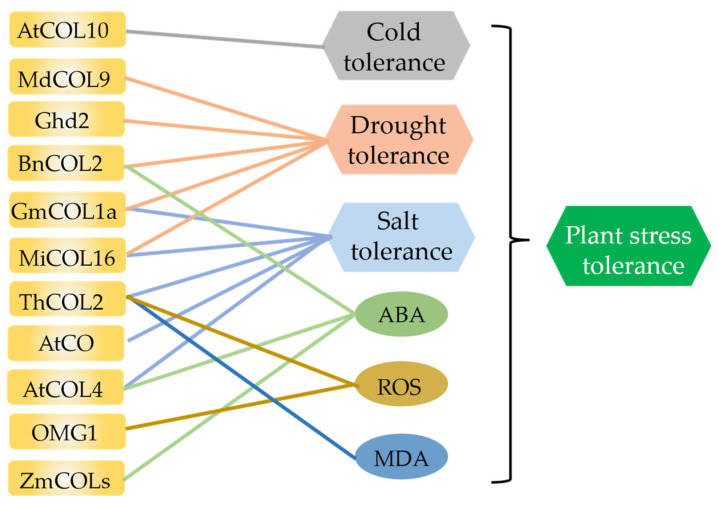
Involvement of COLs from different species in stress responses. Among them, AtCOL10 is involved in cold response; MdCOL9, Ghd2, BnCOL2, GmCOL1a, and MiCOL16 are involved in drought response; GmCOL1a, MiCOL16, ThCOL2, AtCO, and AtCOL4 are involved in salt response; BnCOL2, AtCOL4, ZmCOL, and HaCOL are involved in ABA regulation; and ThCOL2 and OMG1 are involved in ROS regulation. ThCOL2 is also involved in MDA regulation, and indirectly involved in plant stress response.

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
