# Peer review of "Involvement of CONSTANS-like Proteins in Plant Flowering and Abiotic Stress Response"

_ijms, 2023, doi:10.3390/ijms242316585_

Round 1

Reviewer 1 Report

Comments and Suggestions for Authors

The authors have done a good job reviewing the extensive literature about CONSTANS and COL genes in plants. 

The figures are sufficiently clear to represent a very complex regulatory network. I feel the need for an introduction about long day/short day and inductive vs non-inductive photoperiods.

29-32 - this sentence is wordy and not very clear

line 33 - Arabidopsis is with a capital A, line 40 is "arabidopsis". I suggest using "Arabidopsis" with the capital A as the commonly accepted name for A. thaliana. Still, please be consistent.

line 60 - Group I PROTEINS include... also in lines 61 about the other groups.

line 64 - I suggest writing the names of amino acids and not the single-letter code here.

line 66 - demonstrated THAT the B-Box domain

line 73 - usually "import" is used rather than "importation"

3. Assembly of COL Transcription Complex

I recommend reading the Antonio Chaves-Sanjuan et al., 2020 paper "Structural determinants for NF-Y subunit organization and NF-Y/DNA association in plants" to get more insight into the assembly of the NF-Y trimer in complex with its DNA target.

line 94 - missing citation https://pubmed.ncbi.nlm.nih.gov/33098724/

line 115 - I would not define "recent" as a 2014 work. It is 10 years old now. 

line 118 - Figure 1 appears in the proof before being mentioned in the text

section 4.1 Regulation of CO in Photoperiod Flowering

I would rewrite the 143-153 paragraph listing first the transcriptional regulators and then the post-transcriptional regulators. Now it is switching back and forth and it isn't easy to follow. Also, it would be worth specifying the differences between long and short-day conditions in Arabidopsis.

line 159 - GIGANTEA first appears here as an acronym, please specify the full name

line 238 - citation suggested: Beata Siemiatkowska et al., 2022.

https://academic.oup.com/pcp/article/63/9/1285/6647599?login=true#372831123

line 278 - which ABF?

section 5.2 COL participates in abiotic- stress response

It would be important to discuss the fact that most of the COL studies reported involve ectopic overexpression of COL proteins, and the function of most of these genes/proteins is still not very clear.

line 317-319 - sentence not clear

Comments on the Quality of English Language

Generally, the English language could be improved by simply using grammar software.

I have highlighted some errors in the previous section as an example, however, the entire manuscript needs some English editing. Please consider asking a native English speaker for help (which I am not).

Reviewer 2 Report

Comments and Suggestions for Authors

The manuscript is a review on the role of CONSTANS-like proteins in flowering and resistance to abiotic stresses. The authors describe in great detail the role of the COL protein domain and structure of COL transcription complex. The presented review is for a very hermetic group of geneticists. Legends for Figures are not very precise and it is difficult to understand the diagram without carefully reading the text. There is also no preliminary information on: division of plants according to photoperiod requirements, into short-day and long-day plants, absolutely demanding and less demanding plants, and photoperiod-indifferent plants. In short, the physiological background of the flowering process is missing. It is not known whether photoperiod neutral plants have a genetically similar flowering pattern to the one presented by the authors. Similarly, abiotic stresses are in no way described in the context of flowering induction or inhibition. Figure 4 is incomprehensible. For this reason, I suggest supplementing the introduction with the above information.

The legends for the figures should be better described so that the reader can understand from the diagram what the role of the proteins or genes in question is in the flowering process and in connection with photoperiod or stress resistance. A major shortcoming is the way in which the literature is cited. The authors never provide the name of the journal in which a given work was published.

Round 2

Reviewer 2 Report

Comments and Suggestions for Authors

The paper can be published